# Placebo in Functional Neurological Disorders: Promise and Controversy

**DOI:** 10.3390/healthcare13222863

**Published:** 2025-11-11

**Authors:** Natalia Szejko, Ali Abusrair, Tomasz Pasierski, Simon Schmitt, Catharina Cramer, Tomasz Pietrzykowski, Anna Dunalska, Kamila Saramak, Katarzyna Śmiłowska, Tereza Serranova, Kirsten R. Müller-Vahl

**Affiliations:** 1Clinic of Psychiatry, Social Psychiatry and Psychotherapy, Hannover Medical School, 30-625 Hannover, Germany; schmitt.simon@mh-hannover.de (S.S.); cramer.catharina@mh-hannover.de (C.C.); mueller-vahl.kirsten@mh-hannover.de (K.R.M.-V.); 2Department of Bioethics, Medical University of Warsaw, 02091 Warsaw, Poland; tomasz.pasierski@wum.edu.pl; 3Department of Experimental and Clinical Pharmacology, Center for Preclinical Research and Technology CEPT, Medical University of Warsaw, 02089 Warsaw, Poland; 4Child Psychiatry Ward, Babinski Hospital in Lodz, 91-229 Lodz, Poland; 5Neurology Division, Department of Internal Medicine, Qatif Health Network, Qatif 31911, Saudi Arabia; abusrair.md@gmail.com; 6Department of Psychiatry, Psychotherapy and Psychosomatics, RWTH Aachen University, 52-074 Aachen, Germany; 7Department of Law and Administration, University of Silesia in Katowice, 40007 Katowice, Poland; tomasz.pietrzykowski@us.edu.pl; 8Department of Psychiatry, Faculty of Health Sciences, Medical University of Warsaw, 02091 Warsaw, Poland; anna.dunalska@wum.edu.pl; 9Department of Neurology, Medical University of Innsbruck, 6020 Innsbruck, Austria; kamilasaramak@gmail.com; 10Department of Neurology, Regional Specialist Hospital of St. Barbara in Sosnowiec, 41214 Sosnowiec, Poland; kasia.smilowska@gmail.com; 11Department of Neurology and Center of Clinical Neuroscience, First Faculty of Medicine, Charles University and General University Hospital, 12821 Prague, Czech Republic; tereza.serranova@vfn.cz

**Keywords:** placebo effect, nocebo effect, lessebo effect, functional neurological disorders, ethics

## Abstract

Placebo, nocebo, and lessebo effects are very frequent in patients with both neurological and psychiatric disorders. Interestingly, the neural mechanisms underlying placebo effects have been found to be the same as or similar to mechanisms targeted by active pharmaceutical interventions for many of these disorders. In the case of functional neurological disorders (FNDs), there are shared neural substrates between the central nervous system “placebo network” and the dysfunctional networks implicated in the pathophysiology. These networks are primarily involved in emotion regulation, stress responses, and the sense of self-agency. Therefore, placebo effects have also been discussed as therapeutic interventions in FNDs. Such an approach, however, has a variety of ethical implications evolving around informed consent, autonomy, nonmaleficence, beneficence, and justice. In this paper, we discuss the use of placebo, nocebo, and lessebo in FNDs as well as related ethical issues. Overall, the use of placebo in FNDs is currently still considered controversial both for diagnostic as well as therapeutic purposes. Although it is a safe and almost unique intervention, its use violates the core principles of medical ethics and doctor–patient interactions involving autonomy or openness in the therapeutic relationship.

## 1. Introduction

Functional neurological disorders (FNDs) reflect a group of heterogenous disorders in which the primary pathophysiology is alterations in functioning brain circuits rather than structural brain pathologies [1]. It is characterized by internal inconsistency and variability over time, either within or between tasks [1,2,3,4,5]. Symptoms typically change with attentional focus (for example, suppression during distraction), with expectations such as suggestion or placebo, or with illness beliefs. Demonstrating such inconsistency is central to making a positive diagnosis. Although incongruity or incompatibility with recognized neurological disease remains in the diagnostic criteria, this is increasingly viewed as a problematic exclusion criterion, as not all disease patterns are fully understood [6]. It is increasingly conceptualized that FNDs reflect a large-scale, multi-network disorder, involving abnormalities both within and across brain circuits implicated in self-agency, emotion processing, attention, homeostatic regulation, interoception, multimodal integration, and cognitive as well as motor control [7,8,9,10].

FNDs may result in high levels of disability and an impairment of quality of life similar to that found in other neurological conditions [7,8]. FNDs are challenging to manage, with the pathways of diagnosis and specialized multidisciplinary care having only been recently developed [9]. A growing amount of evidence confirms that explanation and education, and physical and occupational therapy and psychotherapy, when appropriate, are the most effective treatment options [9].

A placebo refers to any treatment or an element of treatment that produces effects through psychological or psychophysiological mechanisms. The use of a placebo in a group of patients has been postulated to have both diagnostic and therapeutic values [11,12,13]. Distinguishing between the placebo response and the placebo intervention is critical to understanding their respective scientific and ethical implications [14]. The placebo response reflects psychobiological symptom changes driven by expectation, context, and the therapeutic interaction rather than the treatment’s intrinsic properties [15,16]. In contrast, a placebo intervention involves the intentional clinical use of an inert or non-specific treatment to evoke such effects, raising distinct ethical considerations [14]. The nocebo effect, opposite to the placebo effect, refers to the clinical worsening caused by a drug or inactive substance due to the patient’s negative expectations rather than any pharmacological mechanism [15,17]. Similarly, the lessebo effect describes the reduced drug efficacy in randomized controlled trials (RCTs) when participants’ awareness of possible placebo assignment generates negative expectations about the treatment [15,18].

These effects are believed to be mediated by a variety of overlapping external (e.g., environmental cues and the patient–physician relationship) and internal (e.g., expectations, emotions, and cognitive framing) factors [19]. In particular, it is widely known that elements of the surroundings can influence patients’ expectations and enhance the placebo effect, even if, in fact, no placebo is administered [20,21,22,23,24,25]. Importantly, these cues have an impact on credibility, safety, and efficacy. Important elements of the surroundings include the clinical setting, the appearance of the clinic and the hospital, medical uniform, and other aspects of the medical equipment [26].

However, today it is believed that placebo responses are not limited to psychological factors. The administration of a placebo has a clear modulating effect on neuronal circuits, mainly in the prefrontal cortex, which is implicated in expectation of relief, especially in placebo effects on analgesia [27,28]. Other anatomical circuits involved include the anterior cingulate cortex which contributes to pain modulation [21], emotional regulation, and decision making; the insular cortex implicated in interoception and pain perception [29]; and the nucleus accumbens primarily responsible for reward and motivation [30]. Additional regions include the brainstem [31,32], the amygdala [33], and the hippocampus [34]. From a pathophysiological perspective, mind–body interactions play a central role in the development of the placebo effect [19,24,35]. Overall, two main mechanisms have been proposed: the expectation of therapeutic benefit and the activation of neuronal circuits related to learning and reward pathways shaped by previous therapeutic experiences [36,37].

Much of the attention given to placebo effects in medicine has focused on their role in placebo-controlled clinical trials and the nuisance they can pose on measurements of clinical efficacy [38,39,40]. Overall, placebo interventions could be divided into (i) pure, where an indeed ineffective substance is used such as saline or water (ii) impure, where the substance used is potentially active but not in a particular clinical indication [19].

In the context of neurological disorders, these include patients with Parkinson’s disease (PD), multiple sclerosis (MS), chronic pain, and neuropsychiatric disorders such as Tourette syndrome (TS) [22,41,42,43,44]. Interestingly, for most of these disorders, the same neuronal networks are being activated when active medication or a placebo are being administered [45,46]. Based on current knowledge, in particular, PD and TS can be treated as pathognomonic examples, since high rates of placebo response have been demonstrated: in PD around 16% of patients (with a range reaching up to 55% in some cases) [47,48,49,50,51,52,53] and in TS (with rates 25–35%) [54,55,56].

Despite ongoing ethical debate, many clinicians consider prescribing pure and impure placebos acceptable in specific contexts. Interestingly, patient surveys also reveal a relatively high level of openness toward placebo interventions when framed within an honest and supportive clinical relationship.

The purpose of this narrative review is to present an overview of ethical considerations related to the use of placebo in FNDs by mainly addressing the core principles of justice, informed consent, autonomy, non-maleficence, beneficence, and justice.

## 2. Role of Placebo in FNDs

### 2.1. Shared Mechanisms Between FNDs and Placebo Effect

From pathophysiological perspective, the underlying mechanisms for the development of FNDs are quite complex, which is also reflected in the complexity of biomarkers that have been tested [57]. Recent neurobiological research has highlighted overlapping mechanisms between FNDs and the placebo response [14]. Both phenomena engage brain networks involved in expectancy, self-agency, and emotional regulation, particularly the anterior cingulate cortex, insula, prefrontal cortex, and amygdala (Figure 1). 

Functional imaging studies demonstrate that placebo responses are not “imaginary” but are associated with measurable changes in brain activity and connectivity [58]. Current theoretical frameworks based on predictive coding models of brain function highlight the central role of aberrant motor and sensory predictions driven by abnormally focused attention [59]. Contrary to historical assumptions, individuals with FNDs are not generally more suggestible than healthy controls [59,60]. Taken together, both FND symptoms and placebo effects can be understood within a predictive coding framework, in which expectations shape bodily experiences and motor output. This overlap suggests that the placebo response may provide an experimental model to better understand symptom generation and maintenance in FNDS.

Suggestibility in FNDs is closely linked to the placebo response, as both involve symptom modulation through expectation and contextual cues. Patients with FNDs often exhibit marked responsiveness to suggestion during examination, a phenomenon paralleled by their high rates of improvement with placebo interventions. This overlap highlights how top-down processes including attention, belief, and prior expectation can directly shape motor and sensory output in FNDs. Clinical studies have shown that suggestion-based and placebo responses in FNDs are stronger than in many organic conditions, supporting the view that FNDs represent an exaggerated manifestation of normal brain mechanisms underlying symptom perception and placebo responsiveness [11,60]. However, this does not extend to certain placebo-suggestive procedures, indicating that any heightened suggestibility is restricted to particular contexts and is not broadly diagnostic and/or therapeutic. A summary of the shared mechanisms between placebo, nocebo, and lessebo effects and FNDs is depicted in Table 1. 

### 2.2. Placebo Responsiveness in FNDs

Recent empirical research challenges the longstanding belief that individuals with FNDs exhibit a high susceptibility to placebo effects. In one study using a classic placebo paradigm, participants received a purported anesthetic cream while pain ratings were compared between FND patients and healthy controls [15]. Both groups showed comparable reductions in pain following the placebo application, and conditioning did not further enhance the effect. Notably, a residual analgesic response persisted even after disclosing the placebo nature of the treatment. These results indicate that patients with FNDs may not have heightened placebo responsiveness, although evidence remains limited and based on small samples. Thus, while these findings challenge the notion of general hypersuggestibility in FNDs, further replication in larger and more diverse cohorts is needed.

Further evidence suggests that placebo interventions—particularly those delivered transparently, such as open-label placebos (OLPs)—may hold therapeutic promise in managing FND symptoms [39,40]. In OLP designs, patients are explicitly informed that the treatment is inert yet may still experience improvements when its rationale is explained in a credible and supportive manner. This transparency preserves patient autonomy while leveraging expectancy mechanisms, thereby reconciling ethical integrity with potential clinical benefits. Expanding research on OLPs in FNDs could help clarify how honesty and therapeutic engagement can coexist within evidence-based care.

In parallel, a small feasibility study explored the use of sham transcranial magnetic stimulation as a placebo intervention in individuals with the cognitive subtype of FNDs, in which participants were blinded to the nature of the intervention and informed of the placebo design only during the post-study debriefing. The trial reported subjective improvements in memory and reductions in anxiety, and most participants considered the approach ethically acceptable and justifiable after disclosure [61].

Hypnosis and suggestion, often explored as therapeutic tools in FNDs, are closely linked to placebo mechanisms such as top-down attention, expectancy, and suggestibility. A systematic review of 35 studies (including five RCTs) found that up to 87% of patients experienced clinical improvement with hypnosis or suggestion, though methodological limitations, heterogeneous designs, and variable outcome measures prevent firm recommendations at this stage [62]. The clinical practice literature has also described the “split-screen” technique, in which patients are guided to consciously elicit and suppress their symptoms, thereby learning to exert control through suggestion or self-hypnosis [63,64,65]. Moreover, neuromodulatory interventions such as transcranial magnetic stimulation (TMS) targeting the dorsolateral prefrontal cortex have been shown to increase hypnotizability, potentially enhancing the therapeutic efficacy of hypnosis in FNDs [65]. Moreover, many studies using TMS did not use a stimulation protocol able to elicit durable neuromodulatory changes (single pulses of TMD) and placebo effect can be suspected. Evidence for this provided Garcin et al. who showed the same effect from cortical and spinal root stimulation [66]. Together, these approaches illustrate how hypnosis in FNDs can harness placebo-related processes in a structured and transparent manner, although further controlled trials are needed to establish its clinical role.

### 2.3. Inclusion of Placebo/Nocebo Response in the International Guidelines for Diagnosis and Treatment of FNDs

While placebo mechanisms can influence symptoms in FNDs, their use in clinical practice is limited by ethical and professional boundaries. This balance is reflected in international guidelines, which outline when and how placebo and nocebo effects may be considered in diagnosis and treatment.

In FNDs, changes in expectations play a central mechanistic role, which may provide a rationale for the use of placebo in both diagnosis and treatment [66]. International consensus for FNDs emphasize that diagnosis is based on positive clinical signs and multidisciplinary rehabilitation approaches while explicitly discouraging the use of deceptive placebo interventions as this may lead to misdiagnosis or reinforce stigma by implying that symptoms are “imagined” or even voluntary. Placebo and suggestion are still acknowledged as mechanisms that may contribute to observed improvements in FNDs, particularly in response to interventions such as TMS, but guidelines advise against their intentional therapeutic use based on ethical grounds [67]. Instead, treatment recommendations focus on education, physiotherapy, cognitive behavioral interventions, and speech or occupational therapy delivered within a transparent, collaborative framework [68,69,70]. Similarly to placebo, nocebo response is not formally included as a diagnostic or treatment tool, but guidelines highlight the importance of clinician communication, avoidance of iatrogenic harm, and positive framing of the diagnosis to minimize symptom exacerbation [70]. 

## 3. Ethical Considerations Regarding the Use of Placebo in FNDs

### 3.1. Justice and Resource Allocation

According to the principle of justice, defined by the fair distribution of resources, a number of positive repercussions of placebo administration in FNDs should be taken into consideration. Firstly, it may decrease the health care costs by reducing unnecessary use of therapies not indicated for FNDs [71]. Moreover, administering placebo can be an element of properly designed drug testing. However, the use of placebo is not currently supported by any guidelines or expert opinion and is therefore considered controversial in this group of patients. The diagnosis should not rely solely on a response to placebo, as such responses can also occur in other neurological disorders. On the other hand, establishing the diagnosis often requires access to specialized clinics, which are limited by health care resources and long waiting lists—raising an additional economic argument for the use of placebos in FNDs. Although the application of placebos is considerably cheap, it is limited by the fact that not all clinicians are trained to deliver this effectively, which represents another limitation. Under certain circumstances, it may therefore be a viable option from a health and economic perspective, especially in countries with limited resources, provided there is no risk of improper treatment or no alternative treatment is available.

### 3.2. Beneficence and Non-Maleficence

Another core principle of medical ethics is beneficence, widely corresponding to the promotion of the patient’s wellbeing. By definition, a placebo is usually safe; however, in some patients, a nocebo effect following the administration of an inactive substance could also be observed. If the use of a placebo could indeed shorten diagnostic delays, this would also have beneficial effects for patients as individuals. A key condition for placebo use is the lack of proper treatment available. However, in recent years, there has been a significant paradigm shift in the therapy of FNDs, with real-life, evidence-based data supporting the positive effect of psychoeducation itself, as well as psychotherapy with different modalities tailored to patients’ needs, including cognitive behavioral therapy (CBT), psychodynamic approaches, or hypnotherapy in selected cases. Depending on the clinical presentation, occupational, speech, and physical therapies could also be of benefit. Finally, when underlying depression and/or anxiety or other psychiatric comorbidities such as sleep problems exist and warrant treatment, therapy initiation can also have beneficial effects on FNDs itself.

Related to the principle of beneficence is non-maleficence, corresponding to the avoidance of harm. In this context, the physical harm related to the use of a placebo is minimal. However, there are potentially harmful repercussions from psychological or ethical perspectives with the disruption of therapeutic relationships and the erosion of trust in the treating physician and broader health care system. This is of particular importance as patients with FNDs experience a lot of stigma both from the social perspective, but also even from the health care providers [72]. Still, a significant number of physicians misinterpret or associate FNDs with malingering [73]. For example, a recent survey summarizing knowledge and attitudes towards FNDs among Czech, Slovak, and Italian neurologists [73] showed that only a minority of neurologists considered malingering unlikely. Given these factors, there is a potential for harm and the disruption of trust. In the context of non-maleficence, impure placebos could however cause some side effects and can therefore be more problematic. This is of particular importance as impure placebos constitute the majority of placebos used nowadays. In addition, the use of a placebo could be interpreted as an “easy fix” which does not correspond to current evidence about pathophysiology and efficacious treatments for FNDs. Taken together, placebos should be conceived as an adjunctive therapy, at best. Further criticism of the use of active placebos indicates that it might constitute an excessive and unnecessary medicalization of patients who do not have any underlying somatic pathology. 

### 3.3. Autonomy and Informed Consent

Another crucial ethical principle related to patients’ care is autonomy which corresponds to the rights of patients to make informed decisions about their care [12,13,16]. Again, the application of a placebo in such a case could lead to the disruption of the doctor–patient relationship and the sensation of deception, misinformation, and a lack of transparency. Following this concern, the American Medical Association’s Code of Medical Ethics as well as the British Medical Association forbid the deceptive use of placebos. If, however, the patient is duly informed that the therapeutic effects of an administered medicine are possible but cannot be confirmed, the placebo used under the above defined conditions seems ethically justifiable. 

### 3.4. Ethics of Care and Consequentialism

A complementary approach is offered by the ethics of care, which emphasizes the therapeutic relationship [13,16]. Here, honesty and empathy are themselves therapeutic tools: a transparent discussion of placebo mechanisms can foster trust, engagement, and even enhance placebo responses without deception. Early studies on open-label placebos support this idea, showing that patients can benefit even when they know the intervention is inert, provided it is delivered with a credible rationale and in a supportive context.

## 4. Conclusions and Future Directions

The role of placebos in functional neurological disorders (FNDs) remains complex and ethically debated. While placebo interventions may enhance positive expectations and therapeutic engagement, their use raises concerns about transparency, autonomy, and trust. Deceptive placebo use in FNDs is ethically indefensible, but open-label and suggestibility-based strategies might provide ethically acceptable adjuncts in care. Non-deceptive approaches—such as open-label placebos or strategies that reinforce reversibility and symptom improvement—are viewed as more acceptable. However, placebo responsiveness should not be used diagnostically, as symptom fluctuations in FNDs likely reflect intrinsic variability and top-down modulation rather than pharmacologic effects. 

Beyond clinical care, FNDs carry significant medico-legal and social challenges, including difficulties in obtaining disability or insurance coverage due to misconceptions that symptoms are “not real.” These issues parallel placebo mechanisms, both shaped by expectation, belief, and contextual framing. Clear diagnostic documentation, effective communication, and public awareness are therefore crucial to reduce stigma and ensure equitable support. Future work should focus on structured, ethically grounded expectation-based therapies and greater advocacy to address the societal impact of FNDs.

Future research should clarify the neural and psychological mediators of placebo responsiveness in FNDs, identify predictors of benefit, and evaluate how OLP paradigms can be integrated into multidisciplinary treatment models. Comparative studies examining placebo, nocebo, and expectancy effects could further inform best practices and ethical standards.

## Figures and Tables

**Figure 1 healthcare-13-02863-f001:**
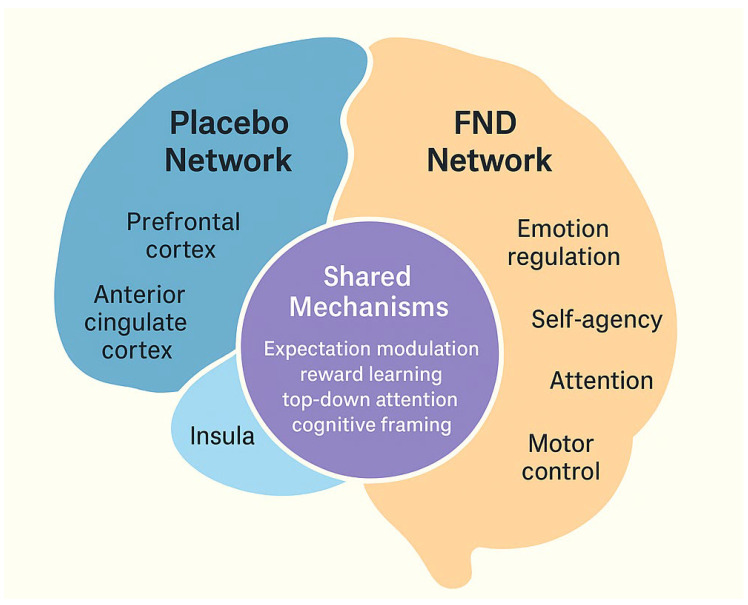
**Integration of placebo mechanisms in functional neurological disorders (FNDs).** The schematic illustrates the overlap between brain networks involved in the placebo response (prefrontal cortex, anterior cingulate cortex, insula, amygdala) and those implicated in FNDs. Both engage circuits related to expectation, attention, and emotion regulation, providing a shared neurobiological basis for therapeutic responsiveness.

**Table 1 healthcare-13-02863-t001:** Comparison of placebo, nocebo, and lessebo effects and their relevance to FNDs.

Effect	Definition	Core Mechanism	Relevance to FNDs
Placebo Effect	Symptom improvement following an inert or non-specific treatment, driven by positive expectations and contextual cues.	Activation of prefrontal, anterior cingulate, and insular networks; modulation of dopaminergic reward pathways.	Central to both FNDs pathophysiology and therapy; may contribute to symptom improvement via expectation and attention modulation.
Nocebo Effect	Worsening of symptoms due to negative expectations or conditioning, independent of pharmacologic action.	Heightened anxiety and anticipatory processing; involvement of limbic and stress-related circuits.	May exacerbate FNDs symptoms through maladaptive expectations or poor communication.
Lessebo Effect	Reduced efficacy of an active treatment due to participants’ awareness of possible placebo assignment in clinical trials.	Negative expectancy and uncertainty reducing engagement of reward and belief circuits.	Less directly applicable; may influence interpretation of trial data involving FNDs interventions.

## Data Availability

Not applicable.

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
