# Peer review of "Placebo in Functional Neurological Disorders: Promise and Controversy"

_healthcare, 2025, doi:10.3390/healthcare13222863_

Round 1

Reviewer 1 Report

Comments and Suggestions for Authors

The manuscript “Placebo in Functional Neurological Disorders: Promise and Controversy” provides a timely and thorough narrative review of the mechanisms, clinical implications, and ethical challenges surrounding placebo, nocebo, and lessebo effects in patients with functional neurological disorders (FND).

The introduction effectively establishes the relevance of FND in neurology and situates placebo effects within both diagnostic and therapeutic debates. The authors successfully integrate epidemiological data, pathophysiological concepts, and the historical controversies around diagnosis. The background discussion of overlapping brain networks (prefrontal cortex, anterior cingulate cortex, insula, amygdala, and reward pathways) provides a solid neuroscientific foundation, and the framing within predictive coding models is conceptually strong and up to date.

The review’s strength lies in its balance between clinical evidence and ethical analysis. The sections on shared mechanisms between placebo and FND, responsiveness to suggestion, and the role of open-label placebo highlight both the opportunities and the limitations of integrating placebo into clinical pathways. Importantly, the authors do not overstate empirical findings, acknowledging the heterogeneity of study designs and the lack of consistent evidence for heightened placebo responsiveness in FND populations.

The ethical discussion is robust, with careful treatment of autonomy, non-maleficence, beneficence, and justice. The authors convincingly argue that deception undermines therapeutic relationships and patient trust, while highlighting the emerging evidence that open-label placebo can be beneficial without violating ethical principles. The integration of Kantian, consequentialist, and care ethics frameworks adds philosophical depth, and the discussion of economic pressures in resource-limited settings broadens the argument’s relevance.

The review also situates placebo responses in FND within the wider neurology field, drawing parallels with Parkinson’s disease and Tourette’s syndrome. This comparative approach strengthens the argument for considering FND as part of a spectrum of conditions in which expectations and contextual framing play key roles. The extension into medico-legal implications—stigma, malingering misconceptions, disability claims—further enhances the manuscript’s applied relevance.

There are, however, some points for refinement. At times, the manuscript repeats concepts (e.g., predictive coding and suggestibility are introduced multiple times), which could be streamlined for conciseness. The conclusion, though comprehensive, could be sharpened to emphasize more clearly the paper’s central stance: that deceptive placebo use in FND is ethically indefensible, but open-label and suggestibility-based strategies might provide ethically acceptable adjuncts in care. A few stylistic and grammatical issues should be corrected for flow and readability, and the manuscript would benefit from a final round of language polishing. Figures or conceptual diagrams could also be considered to illustrate overlapping networks or ethical decision frameworks, improving accessibility for a multidisciplinary readership.

Overall, this is a well-researched and balanced narrative review that integrates neuroscience, clinical data, ethics, and social implications into a cohesive discussion.

Comments on the Quality of English Language

The manuscript is generally well written and communicates complex ideas clearly; however, there are several language and stylistic issues that would benefit from revision. At the sentence level, there are frequent instances of missing or misplaced articles (“the,” “a”), which make some sentences read less fluently. Subject–verb agreement also occasionally slips, for example when singular nouns are followed by plural verbs. Word choice is at times inconsistent, with terms such as “administring” instead of “administering,” “healtcare” instead of “healthcare,” or “intself” instead of “itself,” which should be corrected.

Redundancy is another recurrent issue. Key concepts—such as predictive coding, suggestibility, and the overlap of placebo and FND networks—are explained more than once, sometimes almost verbatim. This repetition interrupts the flow and can be reduced by consolidating explanations into a single, well-developed passage. Similarly, some phrases are wordy and could be streamlined for conciseness; for example, “application of placebo in such case could lead to disruption” could be shortened to “placebo use may disrupt.”

Transitions between sections are sometimes abrupt, particularly when moving from empirical findings to ethical considerations. The addition of linking sentences would help maintain a smoother narrative progression. The conclusion, while comprehensive, would benefit from sharper phrasing to emphasize the authors’ position more directly, as it currently reiterates earlier points without a decisive closing statement.

Finally, minor stylistic adjustments are needed to improve clarity and consistency. For instance, “openniness” should be corrected to “openness,” “shortcomming” to “shortcoming,” and “cheapier” to “cheaper.” Consistent use of technical terms (e.g., “functional neurological disorder” versus “FND”) should also be checked to avoid unnecessary variation.

Author Response

On behalf of my co-authors, I would like to thank you for your thorough review and insightful comments. We carefully considered all suggestions, and to improve the overall quality of the manuscript, we have made several revisions. Below, we provide a detailed, point-by-point response. All changes have been incorporated into the revised version of the manuscript and highlighted in yellow accordingly.

Reviewer 1

Comment 1:
At times, the manuscript repeats concepts (e.g., predictive coding and suggestibility) which could be streamlined for conciseness.

Response:
We carefully reviewed and consolidated overlapping explanations of predictive coding and suggestibility into a single integrated passage in Section 2.1. Redundant material was removed to improve flow and conciseness.

Comment 2:
The conclusion should more clearly emphasize that deceptive placebo use in FND is ethically indefensible, but open-label and suggestibility-based strategies might provide ethically acceptable adjuncts in care.

Response:
We have rewritten the conclusion to explicitly state this key position (Section 4, page 6-7):

Comment 3:

A few stylistic and grammatical issues should be corrected for flow and readability, and the manuscript would benefit from a final round of language polishing.

Response:
The manuscript has been thoroughly proofread to improve language quality and overall readability.

Comment 4:

Figures or conceptual diagrams could also be considered to illustrate overlapping networks or ethical decision frameworks, improving accessibility for a multidisciplinary readership.

Response:
We appreciate this suggestion. An ilustrative figure that outline the overlapping networks waas added to the revised manuscript. Please refer to the uploaded figure for review.

Comments on quality of English language 5:

There are several language and stylistic issues that would benefit from revision. At the sentence level, there are frequent instances of missing or misplaced articles (“the,” “a”), which make some sentences read less fluently.

Subject–verb agreement also occasionally slips, for example when singular nouns are followed by plural verbs. Word choice is at times inconsistent, with terms such as “administring” instead of “administering,” “healtcare” instead of “healthcare,” or “intself” instead of “itself,” which should be corrected.

Redundancy is another recurrent issue. Key concepts—such as predictive coding, suggestibility, and the overlap of placebo and FND networks—are explained more than once, sometimes almost verbatim. This repetition interrupts the flow and can be reduced by consolidating explanations into a single, well-developed passage. Similarly, some phrases are wordy and could be streamlined for conciseness; for example, “application of placebo in such case could lead to disruption” could be shortened to “placebo use may disrupt.”

Transitions between sections are sometimes abrupt, particularly when moving from empirical findings to ethical considerations. The addition of linking sentences would help maintain a smoother narrative progression. The conclusion, while comprehensive, would benefit from sharper phrasing to emphasize the authors’ position more directly, as it currently reiterates earlier points without a decisive closing statement.

Finally, minor stylistic adjustments are needed to improve clarity and consistency. For instance, “openniness” should be corrected to “openness,” “shortcomming” to “shortcoming,” and “cheapier” to “cheaper.” Consistent use of technical terms (e.g., “functional neurological disorder” versus “FND”) should also be checked to avoid unnecessary variation.

Response:
We performed an extensive language review and did our best to correct all the aforementioned grammatical and language errors. We also improved consistency in terminology (e.g., uniform use of “functional neurological disorder (FND)”). We hope that the revised version has enhanced the overall quality, reduced redundancy, and improved the flow.

Reviewer 2 Report

Comments and Suggestions for Authors

General Comments

The manuscript addresses an important and timely topic: the role of placebo, nocebo, and lessebo effects in functional neurological disorders (FND). The focus on both neurobiological underpinnings and ethical considerations is highly relevant and could interest clinicians, neuroscientists, and ethicists.

The review is comprehensive and well referenced, but the narrative could be made clearer with better organization, some condensation, and sharper distinctions between descriptive, mechanistic, and ethical discussions.

Major Points

Clarity and Structure

  • The manuscript is quite dense and occasionally repetitive (e.g., overlapping explanations of predictive coding, placebo mechanisms, and ethical frameworks). Consider streamlining sections to avoid redundancy and ensure a logical flow.
  • Ethical considerations are well developed but could be structured more systematically (e.g., by presenting each principle-autonomy, beneficence, non-maleficence, justice-in clearly separated subsection

Balance of Evidence vs. Interpretation

  • While the paper reviews both supportive and contradictory findings regarding placebo responsiveness in FND, the narrative sometimes gives the impression of stronger consensus than the evidence allows. Clarify where evidence is limited, preliminary, or inconsistent.
  • Some claims (e.g., placebo as a cost-effective diagnostic tool) could be better balanced with counterpoints about risks of misdiagnosis, stigma, or reinforcing misconceptions about FND.

Terminology and Definitions

  • Distinguish more clearly between placebo response as a psychobiological phenomenon and placebo intervention as a clinical strategy.
  • The lessebo effect is mentioned briefly but not developed in relation to FND - either expand on its relevance or remove for conciseness.

Ethical Analysis

  • The ethical discussion is strong but would benefit from a clearer conclusion: under what conditions, if any, might placebo use in FND be ethically permissible?
  • Consider giving more emphasis to open-label placebo research, as this could reconcile efficacy with respect for autonomy.

Conclusions and Future Directions

  • The conclusions section would be stronger if it more clearly delineated clinical implications (what should practitioners do now) versus research directions (what questions need further study).
  • The discussion of psychedelics and hallucinogens as potential future interventions feels speculative and somewhat tangential; either expand with justification or streamline.

Minor Points

  • Several typographical and stylistic issues (e.g., "automony" → autonomy; "healtcare" → healthcare) need correction.
  • Reference formatting should be checked for consistency. Some in-text citations (e.g., double numbering, missing spaces before brackets) could be cleaned up.
  • Ensure uniform use of abbreviations: FND, CBT, TMS, etc.
  • The conflict-of-interest statement is extensive; ensure it is aligned with journal requirements and properly formatted.

Overall Assessment: This is a well-researched, thorough review with high potential impact. With improved structure, clearer distinctions between evidence and interpretation, and a sharper ethical synthesis, it would make a valuable contribution to the literature on FND and placebo.

Author Response

Comment 1:
The manuscript is dense and occasionally repetitive. Please streamline and ensure a logical flow.

Response:
We condensed several sections, particularly those describing predictive coding and placebo mechanisms, to enhance readability and narrative coherence.

Comment 2:
Ethical considerations could be structured more systematically (autonomy, beneficence, non-maleficence, justice).

Response:
We appreciate this comment and have revised Section 3 to explicitly organize ethical considerations under subheadings:

  • 3.1 Justice and Resource Allocation
  • 3.2 Beneficence and Non-Maleficence
  • 3.3 Autonomy and Informed Consent
  • 3.4 Ethics of Care and Consequentialism

This structure provides clearer navigation and aligns with standard bioethical analysis.

Comment 3:
While the paper reviews both supportive and contradictory findings regarding placebo responsiveness in FND, the narrative sometimes gives the impression of stronger consensus than the evidence allows. Clarify where evidence is limited, preliminary, or inconsistent.

Response:
We now explicitly acknowledge areas of limited or inconsistent evidence, ensuring a balanced representation of findings and uncertainties. Kindly refer to the revised manuscript in the section addressing placebo responsiveness in FND

  • Page 4, line 148
  • Page 4, line 164

Also refer to the first paragraph of section 2.3 (Inclusion of placebo/nocebo response in the international guidelines for diagnosis and treatment of FND) in Page 4 line 180, in which we adrssed limited evidance regarding the application of placebo and nocebo in international guidelines.

Comment 4:

Some claims (e.g., placebo as a cost-effective diagnostic tool) could be better balanced with counterpoints about risks of misdiagnosis, stigma, or reinforcing misconceptions about FND.

Response:
We appreciate this comment. Section 2.3 was revised, and we added the “counterpoints” that placebo and other intervention has the risk of misdiagnosis and may even lead to stigma. Please refer to page 5 line 189.

Comment 5:
Distinguish between placebo response (psychobiological phenomenon) and placebo intervention (clinical strategy).

Response:
A definitional paragraph was added to the Introduction clarifying this distinction and its relevance to FND. Please refer to the third paragraph of the intoduction (Page 2, line 61)

Comment 6:
Expand or remove lessebo effect discussion for clarity.

Response:
We retained a concise definition of the lessebo effect and briefly explained its limited relevance to FND, ensuring clarity without unnecessary expansion. Please refer to the third paragraph of the intoduction (Page 2, line 69)

Comment 7:

The ethical discussion is strong but would benefit from a clearer conclusion: under what conditions, if any, might placebo use in FND be ethically permissible?

Response:
We thank you for this observation. The ethical considerations in FND was throughroly revised. Kindly refer to the Ethical considerations regarding the use of placebo in FND in page 5, starting from line 207.

Comment 8:

Consider giving more emphasis to open-label placebo research, as this could reconcile efficacy with respect for autonomy.

Response:

We appreciate this valuable suggestion. We have expanded the discussion of open-label placebo (OLP) research. Kindly refer to the revised manuscript page 5 line 159.

Comment 9:

The discussion of psychedelics and hallucinogens as potential future interventions feels speculative and somewhat tangential; either expand with justification or streamline.

Response:
This section has been removed in the revised version to maintain clarity and relevance.

Comment 10:
Minor typographical and formatting issues.

Response:
All identified errors have been corrected and reference formatting standardized.

Reviewer 3 Report

Comments and Suggestions for Authors

Thank you very much for the invitation to review this manuscript.

The review is overall very interesting and well written. It addresses an important and timely topic concerning placebo, nocebo, and lessebo effects in functional neurological disorders (FND), highlighting their shared neurobiological mechanisms and the related ethical implications. The discussion is clear and balanced, and the manuscript provides valuable insights into the complexity of therapeutic approaches in FND.

Only a few revisions are needed before publication:

Major revision:

I recommend including a summary table clearly outlining the definitions and main differences between placebo, nocebo, and lessebo effects. This would greatly enhance the clarity and didactic value of the paper.

Minor revision:

At line 38, please correct “disorder s(FND)” to “disorders (FND)”.

Overall, this is a very good and engaging review that will be of interest to both clinicians and researchers.

Author Response

Comment 1 (Major):
Include a summary table outlining definitions and main differences between placebo, nocebo, and lessebo effects.

Response:
We have added a new Table 1 titled “Comparison of Placebo, Nocebo, and Lessebo Effects”, summarizing definitions, core mechanisms, and relevance to FND.

Comment 2 (Minor):
Correct “disorder s(FND)” to “disorders (FND).”

Response:
Corrected as suggested.

Round 2

Reviewer 1 Report

Comments and Suggestions for Authors

No further comments.

Reviewer 2 Report

Comments and Suggestions for Authors

I appreciate the thorough revisions made in response to the previous review. The manuscript has improved substantially in clarity, structure, and ethical depth. The narrative now flows more logically, and the ethical discussion is both systematic and well integrated with the empirical sections.

Specific observations:

1. Overall clarity and flow: The manuscript is much more readable and coherent. Redundant or overly dense sections have been effectively condensed, especially in the discussions of predictive coding and placebo mechanisms.

2. Ethical framework: The reorganization of Section 3 into clear subheadings (justice, beneficence, non-maleficence, autonomy, ethics of care/consequentialism) greatly enhances readability and conceptual precision.

3. Balance of evidence: The authors now appropriately acknowledge areas of limited or inconsistent evidence, avoiding overstatement of consensus. The discussion of placebo responsiveness in FND is balanced and transparent.

4. Counterpoints and risks: The addition of remarks regarding potential misdiagnosis, stigma, and ethical risks (page 5, line 189) provides valuable balance and nuance.

5. Conceptual clarity: The distinction between placebo response and placebo intervention in the introduction (p. 2) is helpful and should remain as currently phrased.

6. Open-label placebo discussion: The expanded treatment of open-label placebo research strengthens the ethical argument for transparency and patient autonomy.

7. Ethical permissibility: The ethical analysis and concluding discussion now clearly articulate under what conditions placebo use in FND might be ethically acceptable. The inclusion of different ethical perspectives (Kantian, consequentialist, and care-based) is commendable.

8. Removed speculative content: Deleting the earlier section on psychedelics/hallucinogens improved focus and relevance.

9. Editorial quality: Typographical and formatting inconsistencies appear corrected.

Overall Evaluation

This is now a well-structured, balanced, and thoughtful paper that makes a meaningful contribution to the ethical and clinical discourse on placebo use in FND. The revisions fully address prior reviewer concerns.

Reviewer 3 Report

Comments and Suggestions for Authors

Well done!